# Graph Adversarial Self-Supervised Learning

**Longqi Yang**

Institute for Quantum Information & State Key Laboratory of High Performance Computing,
College of Computer Science and Technology, National University of Defense Technology,
Changsha 410073, China
Defense Innovation Institute, Beijing 100071, China
yanglongqi19@nudt.edu.cn

**Liangliang Zhang**[*]
Institute of Systems Engineering,
AMS, Beijing, China
vermouthlove@hotmail.com

**Wenjing Yang**[*]
Institute for Quantum Information &
State Key Laboratory of High Performance Computing,
College of Computer Science and Technology,
National University of Defense Technology,
Changsha 410073, China
wenjing.yang@nudt.edu.cn

## Abstract

This paper studies a long-standing problem of learning the representations of a whole graph without human supervision. The recent self-supervised learning methods train models to be invariant to the transformations (views) of the inputs. However, designing these views requires the experience of human experts. Inspired by adversarial training, we propose an adversarial self-supervised learning (GASSL) framework for learning unsupervised representations of graph data without any handcrafted views. GASSL automatically generates challenging views by adding perturbations to the input and are adversarially trained with respect to the encoder. Our method optimizes the min-max problem and utilizes a gradient accumulation strategy to accelerate the training process. Experimental on ten graph classification datasets show that the proposed approach is superior to state-of-the-art self-supervised learning baselines, which are competitive with supervised models.

## 1 Introduction

Learning effective representations of graph-structured data plays an essential role in a variety of real-world applications, including social, biological, molecules, and financial networks [1]. Recently, graph neural networks (GNNs) have emerged as powerful architectures for learning and analyzing graph representations [2, 3, 4, 5, 6]. GNNs typically learn graph representations in a supervised or semi-supervised setting. In practice, obtaining a large number of labels is often difficult or even impossible, especially in specific areas that are very costly, such as in biochemistry. The labeled graphs may be limited, while unlabeled graphs are easy to collect. Self-supervised learning utilizing unlabeled data has made significant progress in computer vision [7, 8, 9, 10, 11, 12, 13] and shows great potential in exploring unlabeled data to enhance graph deep learning [14, 15, 16, 17, 18, 19, 20].

Despite their success, existing self-supervised learning methods rely heavily on handcrafted view, where the *view* here refers to human-defined data transformations to preserve the invariance of their intrinsic properties. In recent years, researchers have designed views of graphs from various levels, including nodes dropping, edge perturbation, attribute masking, subgraph [15], and graph diffusion

---

[*]Corresponding authors.

35th Conference on Neural Information Processing Systems (NeurIPS 2021).

[14]. However, the handcrafted views require expert knowledge and trial and error but also do not yield consistent performance gains across multiple tasks [15]. Therefore, how to automatically search for augmentations for graph data remains an open problem.

GNNs are vulnerable to adversarial attacks, as are deep neural networks. Adversarial attacks usually exploit the gradient information to generate imperceptibly small perturbations that alter the model's output. Adding these adversarial samples to the training set, i.e., adversarial training, can improve the neural network to generalize to out-of-distribution samples [21, 22, 23]. Adversarial training usually leads to a trade-off between robustness and generalization. There has been much research on adversarial training for security purposes [24], in particular, *it is still unclear how to combine adversarial training in self-supervised learning of GNNs to improve the classification accuracy.*

In this paper, we are motivated to address the drawbacks mentioned above and propose a self-supervised learning framework to train a graph neural network without any class labels. We refer to this novel adversarial self-supervised learning approach as *Graph Adversarial Self-Supervised Learning* (GASSL). GASSL directly maximizes the similarity of a graph and its perturbed adversarial graph, relying on neither negative pairs nor handcrafted augmented views. In the training phase, we use the gradient accumulation strategy [25, 21] to accelerate the model training. We verify the effectiveness of GASSL on 10 datasets for the graph classification task including the TU datasets [26] and the *Open Graph Benchmark* (OGB) [27]. We conduct extensive experiments across graph datasets by applying classical GNN models (GCN [4] and GIN[5]) as encoders. Our approach automatically generates challenging views to yield performance gains on multiple tasks compared to handcrafted views. The results show that our method outperforms state-of-the-art graph self-supervised learning and is close to the performance of the supervised GNNs.

Our contribution could be summarized as: (1) We propose a self-supervised learning method GASSL for graph representation learning without human supervision. (2) We use adversarial training to automatically generate challenging views for self-supervised learning in place of handcrafted views, which yield performance gains on multiple datasets. (3) We show that GASSL consistently outperforms state-of-the-art self-supervised models with a significant margin in graph classification tasks. When compared to supervised baselines, GASSL performs on par with or superior to the strong baselines.

## 2   Related work

**Graph neural network (GNN)**   GNN is built on graph structures to learn representation vector $\mathbf{H}_v$ for each node $v \in \mathcal{V}$, which are formalized as the following function:

$$\mathbf{H}_v^{(k+1)} = \text{COMBINE}^{(k)} \left( \mathbf{H}_v^{(k)}, \text{AGGREGATE}^{(k)} \left( \left\{ \mathbf{H}_u^{(k)}, \forall u \in \mathcal{N}(v) \right\} \right) \right), \qquad (1)$$

where $\mathbf{H}_v^{(k)}$ is the embedding of node $v$ at the $k$-th layer, $\mathcal{N}(v)$ denotes a neighbor set of node $v$, and $\mathbf{H}_v^{(0)} = \mathbf{X}_v$. COMBINE and AGGREGATE are functions parameterized by neural networks. After $K$ rounds of message passing, we obtain the final-layer node representations. To obtain the representation of the entire graph $\mathbf{h}_\mathcal{G} \in \mathbb{R}^d$, we need the permutation-invariant READOUT function as follows: $\mathbf{h}_\mathcal{G} = \text{READOUT} \left( \{ \mathbf{H}_v \mid v \in \mathcal{V} \} \right)$. Various GNNs have been proposed [4, 28, 5] with various pooling [29, 30, 31], achieving state-of-the-art performance in graph tasks.

**Adversarial robustness**   For adversarial training, more attention is paid to improving model robustness and less on improving generalization performance. For image classification tasks, combining a contrast learning framework with perturbation of the input samples is effective. CLAE [32] employs the Fast Gradient Sign Method (FSGM), and RoCL [33] adopts the Projected Gradient Descent (PGD) method to improve the generalization performance of the model. Concurrent to our work, Tamkin et al. [34] proposed a model-agnostic network (ViewMaker) that perturbs the input by adding an $\ell_p$ constraint to produce useful views and has successful applications on image, speech, and time-series data. Kong et al. [24] proposed to perturb the features of the input nodes of GNN for better generalization performance and utilized a gradient accumulation strategy to accelerate adversarial training in a supervised learning setting. Out of positive view, Hu et al. [35] proposed to directly learn a set of negative adversaries playing against the self-trained representation.

**Graph self-supervised learning**   Self-supervised learning has recently made new advances in graph representation learning, in which contrast learning [36, 17, 15, 14, 37, 38, 39, 40, 41] has

achieved the state-of-the-art performance. Infograph [17] maximizes the mutual information between the graph-level representation and the representations of substructures of different scales (e.g., nodes, edges, triangles). By doing so, the graph-level representations encode aspects of the data shared across different scales of substructures. GraphCL [15] designed four types of graph augmentations to incorporate various priors, including node dropping, edge perturbation, attribute masking, and subgraph. MVGRL [14] utilized graph diffusion for graph augmentation and found no performance gain for more than two views or multi-scales of encoding. GCC [16] performs a random walk with a restart for each node to sample subgraph as augmentation. GRACE [40] adopt two augmentations, including removing edges and node feature masking. For a thorough review, we refer the reader to the recent survey [42]. However, one limitation shared by all these successful approaches is the handcrafted view, which is the primary goal of our GASSL – how to learn a view automatically without resorting to handcrafting or expert domain knowledge.

## 3  Methodology

We now show how to learn the representation of a graph without handcrafted views with domain expert knowledge. Before that, we will briefly introduce adversarial training in supervised learning.

A graph can be represented as $\mathcal{G} = (\mathcal{V}, \mathcal{E})$, where $\mathcal{V}$ is the set of $|\mathcal{V}| = n$ nodes, the adjacency matrix $\mathbf{A} \in \{0, 1\}^{n \times n}$, along with the $c$ dimensional node attribute $\mathbf{X} \in \mathbb{R}^{n \times c}$. Our goal is to learn from multiple graphs in a dataset and predict the property of a single graph $\mathcal{G}$. The learned $d$ dimensional distributed representation $\mathbf{h}_{\mathcal{G}} \in \mathbb{R}^d$ is applied for downstream tasks (e.g. graph classification task).

**Adversarial robustness**   We start with the definition of adversarial attacks under supervised settings. Given a dataset $\mathcal{D} = (X, Y)$, let $x \in X$ and $y \in Y$ denote a training sample and the corresponding label, respectively. Given a supervised learning model $f_\theta : X \to Y$ with parameters $\theta$. Traditional adversarial attacks maximize the loss within a certain radius from the sample as follows:

$$x^{i+1} = \Pi_{B(x,\epsilon)} \left( x^i + \alpha \text{sign} \left( \nabla_{x^i} \mathcal{L}_{\text{CE}} \left( \theta, x^i, y \right) \right) \right) \tag{2}$$

where $B(x, \epsilon)$ is the $\ell_\infty$ norm-ball around $x$ with radius $\epsilon$, and $\Pi$ is the projection function for norm-ball, $\alpha$ is the step size, $i$ is the attack iterations, $\text{sign}(\cdot)$ returns the sign of the vector, $\mathcal{L}_{\text{CE}}$ is the cross-entropy loss for supervised training. The straightforward way to defend against adversarial attacks is to minimize the loss of adversarial samples. Authors [43] proposed to seek to find optimal parameters $\theta^*$ to minimize the maximum risk for any $\delta$ within a norm ball as follows:

$$\min_\theta \mathbb{E}_{(x,y) \sim \mathcal{D}} \left[ \max_{\delta \in B(x,\epsilon)} \mathcal{L}_{\text{CE}}(\theta, x + \delta, y) \right] \tag{3}$$

where $\delta$ is the perturbation of the adversarial example. The conventional adversarial attacks [43, 44] require to have a class label $y \in Y$, which is not applicable to unlabeled data.

### 3.1  Self-supervised learning on graphs

Self-supervised learning typically design pretext tasks to bring different views of the same instance (positive view) closer and push views of different samples (negative view) farther apart. The simple and performant BYOL [9] does not need to maintain negative views explicitly and depends only on positive views. Inspired by BYOL, we propose GASSL framework (Figure 3.1) to learn graph representation. GASSL comprises the two networks: the *teacher* and *student* networks. The two networks shared the same architecture with different parameters. In detail, the teacher network is defined by a set of weights $\theta$, while the student network using a different set of weights $\xi$, which are an exponential moving average of parameters $\theta$. Given a target decay rate $\beta \in [0, 1]$, after each training step, we perform the following update, $\xi \leftarrow \beta\xi + (1 - \beta)\theta$. Note that the predictor $q_\theta$ is only applied to the teacher network to avoid collapse, leading to an asymmetric architecture.

**Encoders.**   In order to learn the graph representation $z$, we used GNN (defined in (2)) following with a two-layer multi-layer perceptron (MLP) as an encoder $f(\cdot)$. Our framework allows various choices of the network architecture without any constraints. We opt for simplicity and adopt the commonly used GCN [4] and GIN [5].

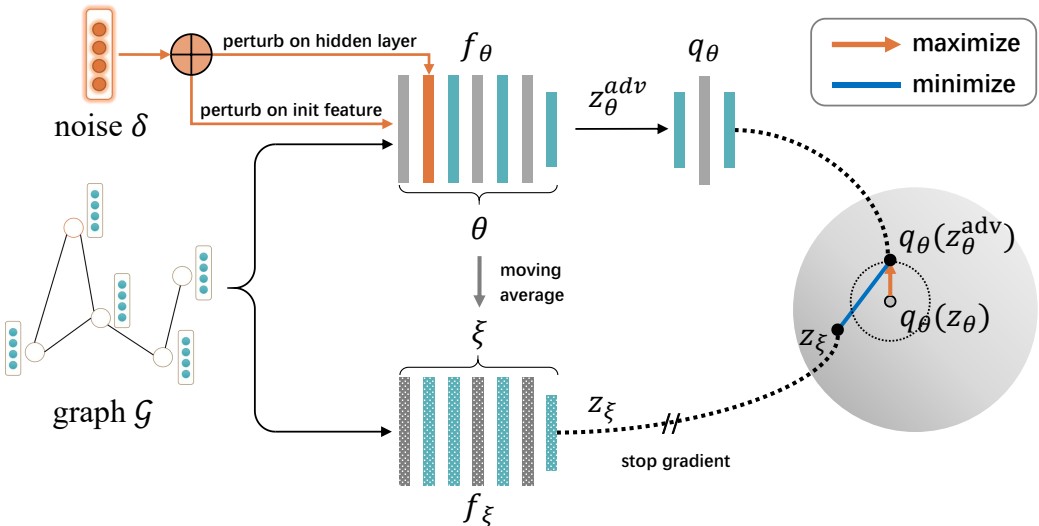

Figure 1: The proposed `GASSL` consists of two networks, called teacher network (upper) and student network (lower). The encoder of the student network is the moving average of the teacher network. The teacher network has an additional MLP $q_\theta$ to avoid collapse. A graph $\mathcal{G}$ is processed by the encoder network ($f_\theta$) and momentum encoder network ($f_\xi$), respectively. `GASSL` alternately performs the outer minimization training and the inner maximization adversarial training.

**Similarity loss** Our method relies on the positive view of the input graph. The traditional way to produce positive view need to be designed manually, such as node dropping, edge perturbation, subgraph. We will introduce to learn a challenging positive view automatically in the next subsection. Here suppose we have obtained the positive view, denoted as $\mathcal{G}'$. We feed $\mathcal{G}'$ and $\mathcal{G}$ to the teacher network and the student network, and obtained the output representations $q_\theta(z'_\theta)$ and $z_\xi$, respectively. We then $\ell_2$-normalized both $q_\theta(z'_\theta)$ and $z_\xi$ to $\bar{q}_\theta(z'_\theta) = q_\theta(z'_\theta)/||q_\theta(z'_\theta)||_2$ and $\bar{z}_\xi = z_\xi/||z_\xi||_2$. We define the mean squared error as follows,

$$\mathcal{L}_{\theta,\xi} = \|\bar{q}_\theta(z'_\theta) - \bar{z}_\xi\|_2^2 = 2 - 2 \cdot \frac{\langle q_\theta(z'_\theta), z_\xi \rangle}{\|q_\theta(z'_\theta)\|_2 \cdot \|z_\xi\|_2} \quad (4)$$

We symmetrize the loss $\mathcal{L}_{\theta,\xi}$ by separately feeding $\mathcal{G}$ and $\mathcal{G}'$ to student network and teacher network to compute $\hat{\mathcal{L}}_{\theta,\xi}$. At each training step, we optimize the loss as follows,

$$\mathcal{L}_{\theta,\xi}^{\texttt{GASSL}} = \mathcal{L}_{\theta,\xi} + \hat{\mathcal{L}}_{\theta,\xi}. \quad (5)$$

### 3.2 Graph adversarial self-supervised learning

To apply adversarial training to self-supervised learning, we optimize the self-supervised loss $\mathcal{L}^{\texttt{GASSL}}$ (Eq 5) to replace the supervised learning cross-entropy loss (Eq 3), allowing adversarial training to produce views without labels.

For self-supervised learning, the generated perturbations should be *challenging* and *faithful* [34]. The generated views should be complex and robust enough for the encoder to produce a useful representation. We generate perturbations by adversarial training to increase the loss between two networks. The perturbation should not make the encoder task impossible. We accomplish this by constraint the radius $\epsilon$ of perturbation. We can add perturbations directly to the input node features or the output of the hidden layer of the GNN encoder.

**Algorithm 1** Graph Adversarial Self-Supervised Learning (`GASSL`)

---

**Input:** Graph $\mathcal{G} = (\mathcal{V}, \mathcal{E})$; input feature matrix $\boldsymbol{X}$; learning rate $\tau$; ascent steps $T$; ascent step size $\alpha$; perturbation bound $\epsilon$, decay rate $\beta$, $\mathcal{L}^{\texttt{GASSL}}(\cdot)$ as objective function.
Initialize $\boldsymbol{\theta}$
**for** $epoch = 1$ **to** $\lceil N_{ep}/T \rceil$ **do**
    $\boldsymbol{\delta}_0 \leftarrow U(-\epsilon, \epsilon)$
    $\boldsymbol{g}_0 \leftarrow 0$
    **for** $t = 1$ **to** $T$ **do**
        $\boldsymbol{g}_t \leftarrow \boldsymbol{g}_{t-1} + \frac{1}{T}\nabla_{\boldsymbol{\theta}}\mathcal{L}^{\texttt{GASSL}}(\mathcal{G}; \boldsymbol{H}^{(1)} + \boldsymbol{\delta}_{t-1})$ `#Accumulate gradient of parameters` $\theta$
        $\boldsymbol{g}_{\boldsymbol{\delta}} \leftarrow \nabla_{\boldsymbol{\delta}}\mathcal{L}^{\texttt{GASSL}}(\mathcal{G}; \boldsymbol{H}^{(1)} + \boldsymbol{\delta}_{t-1})$ `#Update the perturbation` $\delta$ `via gradient ascend`
        $\boldsymbol{\delta}_t \leftarrow \boldsymbol{\delta}_{t-1} + \alpha\boldsymbol{g}_{\boldsymbol{\delta}}/\|\boldsymbol{g}_{\boldsymbol{\delta}}\|_F$
    **end for**
    $\boldsymbol{\theta} \leftarrow \boldsymbol{\theta} - \tau\boldsymbol{g}_T$
    $\boldsymbol{\xi} \leftarrow \beta\boldsymbol{\xi} + (1-\beta)\boldsymbol{\theta}$
**end for**

---

**Perturbation on initial node features** It is common practice to add noise to the node features $\mathbf{X}$, which we denote the perturbed features as $\mathbf{X}^{\text{adv}} = \mathbf{X} + \boldsymbol{\delta}$. The adversarial learning objective following the min-max formulation,

$$\min_{\theta} \mathbb{E}_{\mathcal{G}\sim\mathfrak{G}} \left[ \max_{\|\boldsymbol{\delta}\|_F \leq \epsilon} \mathcal{L}^{\texttt{GASSL}}(\mathcal{G}; \mathbf{X} + \boldsymbol{\delta}) \right] \tag{6}$$

**Perturbation on hidden layers** For GNNs, the features of nodes are aggregated by the neighborhoods. Perturbation on the hidden layer output to affect the nodes with their neighborhoods would produce a more challenging view. We denote the output of first hidden layer as $\mathbf{H}^{(1)}$, the perturbed output is $\mathbf{H}^{(1)} + \boldsymbol{\delta}$. We optimize the following min-max formulation,

$$\min_{\theta} \mathbb{E}_{\mathcal{G}\sim\mathfrak{G}} \left[ \max_{\|\boldsymbol{\delta}\|_F \leq \epsilon} \mathcal{L}^{\texttt{GASSL}}(\mathcal{G}; \mathbf{H}^{(1)} + \boldsymbol{\delta}) \right] \tag{7}$$

The problem (6) and problem (7) are similar, we analyze problem 7 in the following. For problem (7), the outer 'min' of the is non-convex and the inner 'max' is non-concave. This saddle-point problem could be reliably solved with stochastic gradient descent (SGD) for outer minimization and projected gradient descent (PGD) for inner maximization [43]. In this work, we take unbounded adversarial attacks instead. The parameter $\boldsymbol{\delta}$ is updated after each step,

$$\boldsymbol{\delta}_{t+1} = \boldsymbol{\delta}_t + \alpha g(\boldsymbol{\delta}_t)/\|g(\boldsymbol{\delta}_t)\|_F, \tag{8}$$

where $g(\boldsymbol{\delta}_t) = \nabla_{\boldsymbol{\delta}}\mathcal{L}^{\texttt{GASSL}}(\mathcal{G}; \mathbf{H}^{(1)} + \boldsymbol{\delta}_t)$ is the gradient of the loss with respect to $\boldsymbol{\delta}$.

### 3.3 Acceleration training with gradient accumulation

The computation of $\delta$ is inefficient since $T$-step updating takes $T$ forward-backward passes, and the SGD takes only one pass through the neural network. We then leverage the 'free' strategy [25, 21] for efficient adversarial training. The core idea of 'free' strategy is to accumulate gradients of $\nabla_{\boldsymbol{\theta}}\mathcal{L}^{\texttt{GASSL}}$ in each iteration of inner loop and update the model parameter $\boldsymbol{\theta}$ with the accumulated gradients. During the training procedure, suppose we run inner loop $T$ times, each time computing gradient for $\boldsymbol{\delta}_t$ and $\boldsymbol{\theta}_{t-1}$. By taking a decent step along the averaged gradients at $\boldsymbol{H}^{(1)} + \boldsymbol{\delta}_0, \dots, \boldsymbol{H}^{(1)} + \boldsymbol{\delta}_{T-1}$, we approximately optimize the following objective:

$$\min_{\theta} \mathbb{E}_{\mathcal{G}\sim\mathfrak{G}} \left[ \frac{1}{T}\sum_{t=0}^{T-1} \max_{\|\boldsymbol{\delta}_t\|_F \leq \epsilon} \mathcal{L}^{\texttt{GASSL}}\left(\mathcal{G}; \boldsymbol{H}^{(1)} + \boldsymbol{\delta}_t\right) \right] \tag{9}$$

The overall procedure is shown in Algorithm 1.

# 4 Experiment

## 4.1 Datasets

We selected 10 widely used graph classification datasets from TU datasets [26] and Open Graph Benchmark (OGB) [27]. For TU datasets, we select three bioinformatics datasets (MUTAG [45], PTC-MR [46, 45], NCI1 [47]) and three social network datasets (COLLAB [48], IMDB-BINARY [48], IMDB-MULTI [48]). Notably, since the nodes have no features for the social network datasets, we use the one-hot encodings of node degrees as features. We use classification accuracy as an evaluation metric. For OGB datasets, we selected four of the molecular datasets, including HIV, Tox21, ToxCast, and BBBP. We use the ROC-AUC for an evaluation metric. Statistics are reported in Table 1, and more details are described in the Appendix.

Table 1: Statistics of graph classification benchmarks.

| Dataset | MUTAG | PTC-MR | IMDB-B | IMDB-M | COLLAB | NCI1 | HIV | Tox21 | ToxCast | BBBP |
|---|---|---|---|---|---|---|---|---|---|---|
| No. Graphs | 188 | 344 | 1,000 | 1,500 | 5,000 | 4,110 | 41,127 | 7,831 | 8,576 | 2,039 |
| No. Classes | 2 | 2 | 2 | 3 | 3 | 2 | 2 | 12 | 617 | 2 |
| No. Nodes | 17.9 | 25.5 | 19.8 | 13.0 | 74.5 | 29.8 | 25.51 | 18.57 | 18.78 | 24.06 |

## 4.2 Baselines

We select three families of baselines, including graph kernel methods, supervised GNN, unsupervised (self-supervised) methods. The graph kernel methods including shortest path kernel (SP) [49], Graphlet kernel (GK) [50], Weisfeiler-Lehman sub-tree kernel (WL) [51], deep graph kernels (DGK) [48], and multi-scale Laplacian kernel (MLG) [52] reported in [17]. The supervised GNN-based models including GraphSAGE [3], GCN[4], GAT [28], GIN-0 and GIN-$\epsilon$ reported in [5]. In addition, GNN incorporates newly developed pooling methods to further improve performance on graph classification tasks, and we have selected StructPool [29], MinCutPool [30], and Grpah Multiset Transformer (GMT) [31]. The unsupervised methods including random walk [53], node2vec [54], sub2vec [55], and graph2vec [56]. The state-of-the-art self-supervised graph representation learning including InfoGraph [17], MVGRL [14], GraphCL [15], and GCC [16].

## 4.3 Evaluation protocol

For all experiments on the TU dataset, we follow [17, 31] and report the mean 10-fold cross-validation accuracy with standard deviation after 5 runs followed by a linear SVM. The linear classifier is trained using cross-validation on training folds of data, and the best mean classification accuracy is reported. For OGB datasets, we evaluate the performance with their original feature extraction and following the original training/validation/test dataset splits [27]. We train a linear classifier on the top of a frozen encoder on existing self-supervised learning models [7].

We train the model using Adam optimizer with an initial learning rate of $10^{-4}$, and we choose the number of GCN and GIN layers $\in \{2, 3, 4, 5\}$, number of epochs $\in \{20, 40, 100, 200\}$, batch size $\in \{32, 64, 128, 256, 512, 1024\}$, and the SVM parameter $C \in \{10^{-3}, 10^{-2}, \ldots, 10^2, 10^3\}$. The step size $\alpha$ is set to $8 \times 10^{-3}$, the perturbation bound $\epsilon$ is set to $8 \times 10^{-3}$, the embedding dimension is set to 128 (expect HIV set to 512). We also use early stopping with the patience of 20, where we stop training if there is no further improvement on the validation loss during 20 epochs. We conduct all the experiments on an Nvidia TITAN Xp.

## 4.4 The role of adversarial views

**Architecture vs. views** To illustrate that views generated by adversarial training contribute to graph representation learning, we compare with GraphCL [15] that uses handcrafted views. Our proposed GASSL approach differs from the GraphCL in both the self-supervised learning approach and the view generation. Therefore, we construct a new baseline by combining the BYOL architecture with GraphCL's view, named GraphCL-BYOL. We follow the setting of GraphCL and use GIN as the encoder for all comparison methods. From the Table 2, we observe that replacing the backbone of GraphCL from SimCLR to BYOL yields a consistent performance improvement. We choose the

Table 2: Graph classification results (%) on test sets. GraphCL-BYOL indicates that the backbone of GraphCL is replaced with BYOL. The best result is bolded, and the second is underlined.

| | Backbone | View | MUTAG | PTC-MR | IMDB-B | IMDB-M | COLLAB | NCI1 |
|---|---|---|---|---|---|---|---|---|
| GraphCL[15] | SimCLR | Best | 86.8 | – | 71.1 | – | 71.3 | 77.8 |
| | BYOL | NodeDrop | 90.4 | 60.5 | 73.7 | 51.6 | 73.3 | 78.7 |
| | BYOL | EdgePert | 90.5 | 59.9 | 73.9 | 50.8 | 71.2 | 78.5 |
| GraphCL-BYOL | BYOL | Subgraph | 89.1 | 59.8 | 72.9 | 51.2 | 72.4 | 78.5 |
| | BYOL | AttrMask | 89.9 | 59.6 | 73.6 | 50.4 | 71.4 | 79.1 |
| | BYOL | Best | 90.5 | 60.5 | 73.9 | 51.6 | 73.3 | 79.1 |
| GASSL (ours) | BYOL | Adversarial | **90.9** | **64.6** | **74.2** | **51.7** | **78.0** | **80.2** |
| gain from backbone (SimCLR → BYOL) | | | 3.7 | – | 2.8 | – | 2.0 | 1.3 |
| gain from view (GraphCL → Adversarial) | | | 0.4 | 4.1 | 0.3 | 0.1 | 4.7 | 0.9 |

best result as a baseline and compare it with our approach. Our GASSL obtained a boost ranging from $0.1\% \sim 4.7\%$ using adversarial training. It is worth mentioning that compared to GraphCL, our approach improves $6.7\%$ on COLLAB, which backbone contributes $2.0\%$ and view contributes $4.7\%$. The improvement in classification performance indicates the effectiveness of adversarial training.

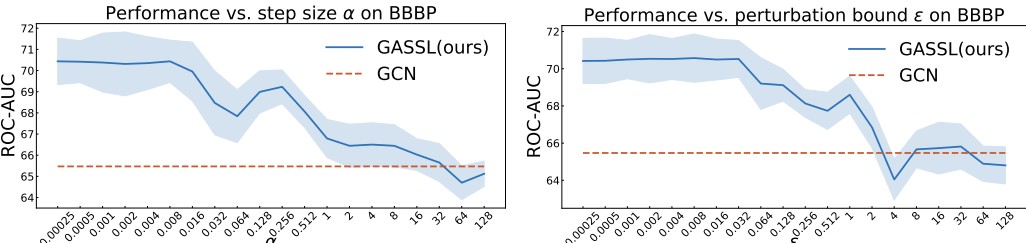

Figure 2: Effects of step size $\alpha$ and perturbation bound $\epsilon$ on BBBP dataset.

**Effect of step size $\alpha$ and perturbation bound $\epsilon$** Step size $\alpha$ and perturbation bound $\epsilon$ are critical factors of adversarial training. We evaluate the effect of $\alpha$ and $\epsilon$ on the classification accuracy of the BBBP dataset. We separately vary $\alpha$ and $\epsilon$ in the range $\{0.00025, 0.0005, \ldots, 64, 128\}$, while fix the other as $0.008$. From Figure 2, the algorithm achieves the best classification performance when $\alpha \leq 0.008$. As the step size increases, the classification performance gradually decreases. Observing the perturbation bound $\epsilon$, we find that the algorithm performance changes similarly. As mentioned in Section 3.2, the perturbations should be challenging and faithful. An overly large step size $\alpha$ or perturbation bound $\epsilon$ leads to perturbed samples that deviate too much from the input graph, and the encoder can hardly learn a useful representation. In the following experiments, we set $\alpha = \epsilon = 0.008$.

## 4.5 Comparison with state-of-the-art

### 4.5.1 Results on TU datasets

The results shown in Table 3 suggest that GASSL achieves state-of-the-art results with respect to unsupervised models. For example, on MUTAG it achieves $90.9\%$ accuracy, a $1.9\%$ relative improvement over the previous state-of-the-art. For kernel methods, our approach achieves better performance on all datasets. When compared to supervised baselines individually, our model outperforms Graph-SAGE in all datasets and outperforms GCN in 4 out of 6 datasets, e.g., a $4.3\%$ relative improvement on GCN for the MUTAG dataset.

Our approach outperforms the state-of-the-art contrastive learning approaches. For example, compared to MVGRL, GASSL has a relative improvement of $2.55\%$ on average across all datasets. GASSL outperforms GraphCL and GCC with a relative improvement of $4.08\%$ and $5.31\%$, respectively.

Table 3: **Graph classification results** on test sets. The reported results are mean and standard deviation over 5 different runs. The compared numbers are from the corresponding papers under the same experiment settings.

| | Dataset | MUTAG | PTC-MR | IMDB-B | IMDB-M | COLLAB | NCI1 |
|---|---|---|---|---|---|---|---|
| Kernel | SP ([49]) | $85.2 \pm 2.4$ | $58.2 \pm 2.4$ | $55.6 \pm 0.2$ | $38.0 \pm 0.3$ | – | – |
| | GK ([50]) | $81.7 \pm 2.1$ | $57.3 \pm 1.4$ | $65.9 \pm 1.0$ | $43.9 \pm 0.4$ | $72.8 \pm 0.3$ | $62.3 \pm 0.3$ |
| | WL ([51]) | $80.7 \pm 3.0$ | $58.0 \pm 0.5$ | $\mathbf{72.3 \pm 3.4}$ | $\mathbf{47.0 \pm 0.5}$ | – | $\mathbf{80.0 \pm 0.5}$ |
| | DGK ([48]) | $87.4 \pm 2.7$ | $60.1 \pm 2.6$ | $67.0 \pm 0.6$ | $44.6 \pm 0.5$ | $\mathbf{73.1 \pm 0.3}$ | $62.5 \pm 0.3$ |
| | MLG ([52]) | $\mathbf{87.9 \pm 1.6}$ | $\mathbf{63.3 \pm 1.5}$ | $66.6 \pm 0.3$ | $41.2 \pm 0.0$ | – | – |
| supervised | GraphSAGE([3]) | $85.1 \pm 7.6$ | $63.9 \pm 7.7$ | $72.3 \pm 5.3$ | $50.9 \pm 2.2$ | – | $77.7 \pm 1.5$ |
| | GCN ([4]) | $85.6 \pm 5.8$ | $64.2 \pm 4.3$ | $74.0 \pm 3.4$ | $51.9 \pm 3.8$ | $79.0 \pm 1.8$ | $80.2 \pm 2.0$ |
| | GIN-0 ([5]) | $\mathbf{89.4 \pm 5.6}$ | $64.6 \pm 7.0$ | $\mathbf{75.1 \pm 5.1}$ | $\mathbf{52.3 \pm 2.8}$ | $\mathbf{80.2 \pm 1.9}$ | $\mathbf{82.7 \pm 1.7}$ |
| | GIN-$\epsilon$ ([5]) | $89.0 \pm 6.0$ | $63.7 \pm 8.2$ | $74.3 \pm 5.1$ | $52.1 \pm 3.6$ | $80.1 \pm 1.9$ | $\underline{\mathbf{82.7 \pm 1.6}}$ |
| | GAT ([28]) | $\mathbf{89.4 \pm 6.1}$ | $\mathbf{66.7 \pm 5.1}$ | $70.5 \pm 2.3$ | $47.8 \pm 3.1$ | – | – |
| unsupervised | Random Walk ([53]) | $83.7 \pm 1.5$ | $57.9 \pm 1.3$ | $50.7 \pm 0.3$ | $34.7 \pm 0.2$ | – | – |
| | node2vec ([54]) | $72.6 \pm 10.2$ | $58.6 \pm 8.0$ | – | – | – | $54.9 \pm 1.6$ |
| | sub2vec ([55]) | $61.1 \pm 15.8$ | $60.0 \pm 6.4$ | $55.3 \pm 1.5$ | $36.7 \pm 0.8$ | – | $52.8 \pm 1.5$ |
| | graph2vec ([56]) | $83.2 \pm 9.6$ | $60.2 \pm 6.9$ | $71.1 \pm 0.5$ | $50.4 \pm 0.9$ | – | $73.2 \pm 1.8$ |
| | InfoGraph ([17]) | $89.0 \pm 1.1$ | $61.7 \pm 1.4$ | $73.0 \pm 0.9$ | $49.7 \pm 0.5$ | $70.6 \pm 1.1$ | $73.8 \pm 0.7$ |
| | MVGRL ([14]) | $89.7 \pm 1.1$ | $62.5 \pm 1.7$ | $\mathbf{74.2 \pm 0.7}$ | $51.2 \pm 0.5$ | $71.3 \pm 1.2$ | $75.0 \pm 0.7$ |
| | GraphCL ([15]) | $86.8 \pm 1.3$ | – | $71.1 \pm 0.4$ | – | $71.3 \pm 1.1$ | $77.8 \pm 0.4$ |
| | GCC ([16]) | $86.4 \pm 0.5$ | $58.4 \pm 1.2$ | $71.9 \pm 0.5$ | $48.9 \pm 0.8$ | $75.2 \pm 0.3$ | $66.9 \pm 0.2$ |
| | GASSL-GCN (ours) | $90.4 \pm 7.9$ | $62.2 \pm 6.0$ | $72.7 \pm 0.7$ | $49.6 \pm 2.3$ | $\mathbf{77.9 \pm 2.0}$ | $77.0 \pm 1.9$ |
| | GASSL-GIN (ours) | $\underline{\mathbf{90.9 \pm 7.9}}$ | $\mathbf{64.6 \pm 6.1}$ | $\mathbf{74.2 \pm 0.5}$ | $\mathbf{51.7 \pm 2.5}$ | $\mathbf{78.0 \pm 2.0}$ | $\mathbf{80.2 \pm 1.9}$ |

Table 4: **Graph classification results** on test sets. The reported results are mean and standard deviation over five different runs. The compared numbers are from the corresponding papers under the same experiment settings. The encoder uses GCN combined with sum pooling, and GASSL-H and GASSL-X denote perturbation at the encoder's first hidden layer and input layer, respectively.

| Dataset | HIV | Tox21 | ToxCast | BBBP |
|---|---|---|---|---|
| GCN[4] | $76.81 \pm 1.01$ | $75.04 \pm 0.80$ | $60.63 \pm 0.51$ | $65.47 \pm 1.73$ |
| GIN[5] | $75.95 \pm 1.35$ | $73.27 \pm 0.84$ | $60.83 \pm 0.46$ | $67.65 \pm 3.00$ |
| StructPool[29] | $75.85 \pm 1.81$ | $75.43 \pm 0.79$ | $62.17 \pm 1.61$ | $67.01 \pm 2.65$ |
| MinCutPool[30] | $75.37 \pm 2.05$ | $75.11 \pm 0.69$ | $62.48 \pm 1.33$ | $65.97 \pm 1.13$ |
| GMT[31] | $77.56 \pm 1.25$ | $\mathbf{77.30 \pm 0.59}$ | $\mathbf{65.44 \pm 0.58}$ | $68.31 \pm 1.62$ |
| GASSL-X (ours) | $\mathbf{78.67 \pm 1.23}$ | $74.60 \pm 0.76$ | $61.72 \pm 0.34$ | $\mathbf{70.46 \pm 1.21}$ |
| GASSL-H (ours) | $\mathbf{78.68 \pm 1.16}$ | $74.59 \pm 0.81$ | $61.96 \pm 0.55$ | $\mathbf{70.57 \pm 1.25}$ |

### 4.5.2 Results on OGB datasets

We evaluate our method GASSL on 4 OGB datasets. From Table 4, we observed that perturbing at the first hidden layer (GASSL-H) yields a slight performance gain compared to perturbing at the input node features (GASSL-X). Our method outperforms GCN and GIN on all datasets, demonstrating our method's potential to outperform supervised learning on larger datasets. Compared with the stronger baselines like structurePool and MinCutPool, which exploit the graph structure information. For StructPool, GASSL has a 3% and 3.5% gain for the HIV and BBBP datasets, respectively. GASSL outperforms MinCutPool by 3.2% and 4.5% for the HIV and BBBP datasets, respectively. The performance is similar on the Tox21 and ToxCast datasets. Our GASSL performs inferior to GMT on Tox21 and ToxCast and superior to GMT on HIV and BBBP. The above results show that our GASSL method can learn a good representation of the graph and outperforms even the state-of-the-art supervised learning methods.

### 4.6 Ablation studies

**Effect of batch size** We analyze the sensitivity of the algorithm to the batch size on four OGB datasets. We selected batch size from $\{32, 64, 128, 256, 512, 1024\}$. From the Table 5, we observe

Table 5: Effect of batch size on the test ROC-AUC (%) on four OGB datasets, with GCN as the encoder. X and H denoting perturbation on the input layer and the first hidden layer, respectively.

| Dataset | HIV | | Tox21 | | ToxCast | | BBBP | |
|---|---|---|---|---|---|---|---|---|
| Batch size | X | H | X | H | X | H | X | H |
| 32 | 77.6 | 77.6 | **74.6** | 74.2 | 60.8 | **62.0** | **66.5** | 68.9 |
| 64 | 77.1 | 77.4 | 72.3 | **74.6** | 61.0 | 61.9 | 65.3 | 69.5 |
| 128 | **78.7** | **78.7** | 73.6 | 73.3 | 60.4 | 60.9 | 66.1 | **70.6** |
| 256 | 75.5 | 77.0 | 71.9 | 73.0 | 60.9 | 61.2 | 66.0 | 67.9 |
| 512 | 75.9 | 76.5 | 71.1 | 72.3 | **61.3** | 60.6 | 65.5 | 67.2 |
| 1024 | 75.2 | 75.1 | 70.9 | 71.5 | 61.2 | 59.6 | 61.2 | 65.4 |
| Average | 76.7 | **76.8**(+0.1) | 72.4 | **73.2**(+0.8) | 60.9 | **61.0**(+0.1) | 65.1 | **67.1**(+2.0) |

that `GASSL` performs stably under different batch sizes. The performance gradually decreases as the batch size increases. In particular, `GASSL` performs well for a batch size of 128. The benefit is that `GASSL` can be trained with fewer resources. Moreover, the perturbation on the first hidden layer output consistently leads to better test accuracy than perturbation on the input node features.

Table 6: Effect of the number of GNN layers and embedding dimension on the test ROC-AUC (%) on four OGB datasets, with GCN as encoder.

| Layers | HIV | Tox21 | ToxCast | BBBP | Dimension | HIV | Tox21 | ToxCast | BBBP |
|---|---|---|---|---|---|---|---|---|---|
| 2 | **78.7** | **73.3** | **60.9** | **70.6** | 128 | 74.8 | 73.3 | 60.9 | **70.6** |
| 3 | 73.8 | 72.2 | 60.3 | 69.6 | 256 | 75.5 | **73.6** | 61.0 | 66.5 |
| 4 | 72.1 | 72.5 | 59.2 | 63.0 | 512 | **78.7** | 72.2 | **61.1** | 63.8 |
| 5 | 71.8 | 72.3 | 60.5 | 63.8 | 1024 | 77.2 | 72.1 | 60.0 | 66.9 |

**Effect of the number of GNN layers**   We evaluate the effect of the number of layers on the classification accuracy using ROC-AUC performance on the OGB dataset, using GCN as the encoder, and selecting the number of layers from $\{2, 3, 4, 5\}$ respectively. From Table 6, we can observe that the performance of `GASSL` gradually decreases as the number of layers increases, while the best performance is obtained when using a 2-layer encoder.

**Effect of embedding dimension**   We test the effect of encoding dimensions on classification accuracy on the OGB datasets. We choose the best encoding dimension among $\{128, 256, 512, 1024\}$. From Table 6 we observe that for HIV, the test accuracy increases as the encoding dimension increases. We set it to $512$ for HIV and $128$ for the rest, considering the computational efficiency.

Table 7: Effect of ascent steps $T$ on the accuracy and training cost (in seconds) for 200 epochs on MUTAG and IMDB-MULTI datasets.

| Encoder | T | MUTAG | Cost(s) | Speed-up | IMDB-M | Cost(s) | Speed-up |
|---|---|---|---|---|---|---|---|
| | 1 | $89.8 \pm 5.8$ | 37 | 1x | $51.5 \pm 2.3$ | 310 | 1x |
| GIN | 2 | $89.3 \pm 7.1$ | 34 | 1.08x | $51.2 \pm 2.2$ | 231 | 1.34x |
| | 3 | $\mathbf{90.9 \pm 7.9}$ | 30 | 1.23x | $\mathbf{51.7 \pm 2.5}$ | 211 | 1.46x |

**Effect of ascent steps**   We explored the impact of ascent steps $T$ on the performance of graph classification accuracy. We train the model in the same setting and vary $T \in \{1, 2, 3\}$. From Table 7, we observed that our method achieves a stable performance on test accuracy. When $T = 3$, for the IMDB-M dataset, there is an improvement in test accuracy along with a speedup of nearly $1.5$ times. The results for other datasets are similar and are detailed in the Appendix.

# 5    Conclusion and Future work

In this paper, we explore a novel problem of how to learn graph representations without human supervision. We propose an adversarial self-supervised learning framework (GASSL) that automatically generates views using adversarial training. Our approach adversarially generates challenging views to train a self-supervised model. We obtain performance gain by generating views through adversarial training compared to handcrafted views. We use a gradient accumulation training method to improve the training efficiency. We conduct extensive experiments on ten datasets. The results show that our method outperforms state-of-the-art graph self-supervised learning and is close to the performance of the supervised GNNs.

*For potential negative societal impact*, the graph representations can be extended to many fields, such as financial networks, molecular biology. The use of transformations generated by adversarial perturbations does not certainly produce meaningful views. Expert knowledge is also required for domain-specific applications. The *limitation* of our approach falls in that it exploits the uniformly norm-bounded perturbation and ignores the distribution of the data. Besides, taking full advantage of the existing expert knowledge is the potential to improve performance.

In the future, we will explore the following directions: (1) Explore how to effectively combine adversarial training with existing handcrafted views to enhance performance further. (2) Theoretically analyze the use of adversarial training to improve the performance of downstream tasks. (3) Explore non-uniform norm-bounded perturbations on the graph to generate adversarial samples.

## Acknowledgments and Disclosure of Funding

This work was partially supported by the National Natural Science Foundation of China (No. 91948303-1, No. 61803375, No. 12002380, No. 62106278, No. 62101575, No. 61906210, No. 91648204) and the National University of Defense Technology Foundation (No. ZK20-09, No. ZK20-52). We would like to thank the anonymous reviewers for their valuable suggestions.

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
