# OpenReview forum: "Graph Adversarial Self-Supervised Learning"
_NeurIPS.cc/2021/Conference — NeurIPS 2021 Poster_

### Official Review · Reviewer_nLuy · 2021-07-15

**Rating:** 7
**Confidence:** 4

**Summary:**

This paper presents a self-supervised learning method GASSL for graph representation learning. The authors have designed 2 key points:  a) Utilizing BYOL framework to learn graph representation; b) Using adversarial training to automatically generate challenging views for self-supervised learning. The idea seems to be simple but effective. Sufficient experimental analyses are conducted.

**Limitations And Societal Impact:**

Please see the weakness part above.

**Main Review:**

Strengths:

1.	To the best of my knowledge, it is the first time that BYOL has been applied for graph self-supervised learning. While this point seems minor, its benefit is greatly supported by the experiments in Table 2 compared to SimCLR.

2.	Different from Euclidean data like images, how to generate valuable positive samples for graphs is non-trivial and central in the domain. Traditional methods like GCC and GraphCL resort to hand-crafted sampling which easily incurs bias during training. This paper proposes to apply adversarial noise to the input features or hidden layers to generate worse-cast positive samples, with the goal of enhancing robustness for BYOL training. The idea is simple yet interesting.

3.	Experimental evaluations are good on the conducted benchmarks and sufficiently support the claims the authors have made.


Weaknesses

1.	This paper is almost well written. Yet there still remains certain unclarity. In line 123-124, the authors mentioned that “Our method relies on the positive view of the input graph, which commonly contains: node dropping, edge perturbation, subgraph, and these need to be designed manually.” This statement is quite confusing. Does this mean that the proposed adversarial technique is not self-contained and is performed on the perturbed samples generated by existing data augmentation methods? If so, why has the claim “We use adversarial training to automatically generate challenging views for self-supervised learning in place of handcrafted views, which yield performance gains on multiple datasets.”

2.	The authors mentioned that the experiments are conducted on large-scale OGB, which is over-exaggerated. It is true that OGB contains large datasets for benchmarking, but the datasets that applied in this paper are small. Please correct the sentence to make it more professional.

3.	It is well known that involving adversarial objective will hinder the training dynamics. Have the authors met this issue and how to address it if so. It is suggested to provide the convergence curves for better clarification.

4.	Regarding the comparison between initial Perturbation and hidden Perturbation, what about the ablation by considering them both? And what about performing perturbation on other layers (not just the first hidden layer)?

5.	Several important references on graph self-supervised learning are not discussed and cited. See below.


[A] Strategies for pre training graph neural networks, ICLR 2020.
[B] Self-Supervised Graph Transformer on Large-Scale Molecular Data, NIPS 2020.
[C] Graph Representation Learning via Graphical Mutual Information Maximization, WWW 2020.


##########Post Rebuttal###########
After reading the authors' responses carefully (including their replies to my further questions), my concerns are well addressed. I believe the contribution by this paper is novel and valuable, and it does address a crucial issue in graph self-supervised learning. In terms of the writing, I strongly suggest the authors to keep their words and revise their paper according to their responses.

Overall, I decide to increase my score.

**Time Spent Reviewing:**

~5h

---

> ### Author Response · Authors · 2021-08-10
> **Response to Reviewer nLuy**
>
> Thank you very much for your positive and constructive comments.
> We are glad you point out that our approach is simple and interesting and is the first application of BYOL to graph self-supervised learning.
> Below please find the responses to some specific comments.
>
> ----
>
>
> Q1. This paper is almost well written. Yet there still remains certain unclarity. In line 123-124, the authors mentioned that “Our method relies on the positive view of the input graph, which commonly contains: node dropping, edge perturbation, subgraph, and these need to be designed manually.” This statement is quite confusing. Does this mean that the proposed adversarial technique is not self-contained and is performed on the perturbed samples generated by existing data augmentation methods? If so, why has the claim “We use adversarial training to automatically generate challenging views for self-supervised learning in place of handcrafted views, which yield performance gains on multiple datasets.”
>
> A1. We are sorry for the confusion.
> Our approach only uses the view generated by the adversarial training as a positive view.
>
> We have rewritten this paragraph as follows.
>
> >Our method relies on the positive view of the input graph. The traditional way to produce positive view need to be designed manually, such as node dropping, edge perturbation, subgraph. We will introduce to learn a challenging positive view automatically in the next subsection. Here suppose we have obtained the positive view, denoted as G'.
>
>
> Q2. The authors mentioned that the experiments are conducted on large-scale OGB, which is over-exaggerated. It is true that OGB contains large datasets for benchmarking, but the datasets that applied in this paper are small. Please correct the sentence to make it more professional.
>
> A2. Thank you very much for your suggestion.
> We will revise the sentence to make it more accurate.
>
> >The experiments are conducted on Open Graph Benchmark (OGB).
>
> Besides, we will test the proposed approach (GASSL) on larger datasets.
>
> ----
>
> Q3. It is well known that involving adversarial objective will hinder the training dynamics. Have the authors met this issue and how to address it if so. It is suggested to provide the convergence curves for better clarification.
>
> A3.  Our proposed GASSL method converges well on several datasets.
> When the training process is not smooth, it helps to reduce the learning rate appropriately.
> As we cannot submit images here, we will add the convergence curves in the final version.
>
> ----
>
> Q4. Regarding the comparison between initial Perturbation and hidden Perturbation, what about the ablation by considering them both? And what about performing perturbation on other layers (not just the first hidden layer)?
>
> A4. Perturbations can be added to feature X in the input layer or in the hidden layer.
> We chose to apply the perturbation at the first hidden layer.
>
> - In Table 6 on page 9, we experiment with GNN encoders for 2-5 layers and show that using a 2-layer encoder works best. Using a 2-layer encoder requires less memory space.
>
> - We have also conducted experiments for the locations where perturbations are applied on the 4 OGB datasets. The results are shown in the table below. The performance of our algorithm is better when using the 2-layer encoder. Our method achieved the best empirical performance by applying the perturbation at the first hidden layer.
>
> - As the number of layers increases, we observe a decreasing trend in the algorithm's performance. A possible reason is the oversmoothing of the GCN, which causes the representation of the nodes to converge to the same.
>
> - The average number of nodes for the four OGB datasets was 25.51, 18.57, 18.78, and 24.06, respectively. GCN might be more oversmoothed on small-scale graphs.
>
> X: input feature
>
> H0: the first hidden layer
>
> H1: the second hidden layer
>
> Dataset: molHIV
>
> | #layers | perturbed | ROC-AUC(%) | #layers | perturbed | ROC-AUC(%) | #layers | perturbed | ROC-AUC(%) |
> | ---- | ---- | ---- | ---- | ---- | ---- | ---- | ---- | ---- |
> | 2    | X    | **78.7** |  3   | X    | 71.0 |  4   | X    | 72.3 |
> | 2    | H0   | **78.7** |  3   | H0   | 75.8 |  4   | H0   | 72.4 |
> | 2    | H1   | 74.6 |  3   | H1   | 76.0 |  4   | H1   | 73.2 |
> |      |      |      |  3   | H2   | 75.4 |  4   | H2   | 73.9 |
> |      |      |      |      |      |      |  4   | H3   | 71.4 |
>
> Dataset: molTox21
>
> | #layers | perturbed | ROC-AUC(%) | #layers | perturbed | ROC-AUC(%) | #layers | perturbed | ROC-AUC(%) |
> | ---- | ---- | ---- | ---- | ---- | ---- | ---- | ---- | ---- |
> | 2    | X    | 74.6 |  3   | X    | 73.6 |  4   | X    | 73.6 |
> | 2    | H0   | **74.6** |  3   | H0   | 73.2 |  4   | H0   | 74.0 |
> | 2    | H1   | 73.3 |  3   | H1   | 73.1 |  4   | H1   | 73.4 |
> |      |      |      |  3   | H2   | 72.7 |  4   | H2   | 72.8 |
> |      |      |      |      |      |      |  4   | H3   | 73.1 |
>
>
> Dataset: molToxCast
>
> | #layers | perturbed | ROC-AUC(%) | #layers | perturbed | ROC-AUC(%) | #layers | perturbed | ROC-AUC(%) |
> | ---- | ---- | ---- | ---- | ---- | ---- | ---- | ---- | ---- |
> | 2    | X    | 61.7 |  3   | X    | 59.9 |  4   | X    | 60.7 |
> | 2    | H0   | **61.9** |  3   | H0   | 60.5 |  4   | H0   | 60.9 |
> | 2    | H1   | 60.4 |  3   | H1   | 59.9 |  4   | H1   | 60.8 |
> |      |      |      |  3   | H2   | 59.6 |  4   | H2   | 60.2 |
> |      |      |      |      |      |      |  4   | H3   | 59.7 |
>
>
>
> Dataset: molBBBP
>
> | #layers | perturbed | ROC-AUC(%) | #layers | perturbed | ROC-AUC(%) | #layers | perturbed | ROC-AUC(%) |
> | ---- | ---- | ---- | ---- | ---- | ---- | ---- | ---- | ---- |
> | 2    | X    | 70.5 |  3   | X    | 67.1 |  4   | X    | 67.0 |
> | 2    | H0   | 70.6 |  3   | H0   | 69.6 |  4   | H0   | 62.9 |
> | 2    | H1   | **71.3** |  3   | H1   | 69.8 |  4   | H1   | 62.4 |
> |      |      |      |  3   | H2   | 69.8 |  4   | H2   | 62.1 |
> |      |      |      |      |      |      |  4   | H3   | 62.3 |
>
>
> ----
>
>
>
>
>
> Q5.Several important references on graph self-supervised learning are not discussed and cited. See below.
>
> [A] Strategies for pre training graph neural networks, ICLR 2020.
> [B] Self-Supervised Graph Transformer on Large-Scale Molecular Data, NIPS 2020.
> [C] Graph Representation Learning via Graphical Mutual Information Maximization, WWW 2020.
>
> A5. We apologize for the missing of important references.
> We will cite and discuss the above papers in related work.
>
> ----
>
> Finally, thank you very much for your time and constructive comments.
> Your comments have helped dramatically in improving the quality of this article.

---

> > ### Comment · Reviewer_nLuy · 2021-08-18
> > **Further questions.**
> >
> > Thanks for the authors' detailed responses. There are still some concerns:
> >
> > 1. You said you have tested the proposed approach (GASSL) on larger datasets in A2. But I can not find the results in the rebuttal.
> >
> > 2. Regarding Q3, would you please provide external link to show the curves? For example the anonymous github repository.
> >
> > Thanks.

---

> > > ### Author Response · Authors · 2021-08-19
> > > **Response to the further questions.**
> > >
> > > Thank you very much for your comments.
> > >
> > > ----
> > >
> > > 1. We are sorry for the misunderstanding.
> > >
> > > In our previous rebuttal A2, we mentioned
> > > >"Besides, we will test the proposed approach (GASSL) on larger datasets."
> > >
> > > We are looking into improving the efficiency of the algorithm as future work, which is not yet completed.
> > >
> > > ----
> > >
> > > 2. Following your suggestion, we uploaded the training loss curves of 10 datasets to anonymous Github repository.
> > >
> > >       https://github.com/Anonymous20201007/losscurves
> > >
> > > We train the model using Adam optimizer with a learning rate of 1e-4, with GCN as the encoder. For fair comparison, we fix the epoch=200, batch=128, embedding dimension=128. From the figures, we can observe that our approach (GASSL) converges well.
> > >
> > > ----
> > >
> > > Finally, thank you for your constructive comments.

---

> > > > ### Comment · Reviewer_nLuy · 2021-08-19
> > > > **Thank you**
> > > >
> > > > Thank you, the results make sense. I have changed my score.

---

> > > > > ### Author Response · Authors · 2021-08-28
> > > > > **Thank you.**
> > > > >
> > > > > We sincerely thank the reviewer for positive and contrastive comments to strengthen our paper.

---

### Official Review · Reviewer_ry4V · 2021-07-16

**Rating:** 7
**Confidence:** 4

**Summary:**

This paper introduces a novel way to improve unsupervised graph representation learning. The proposed method utilizes the adversarial attacking techniques with a teacher and a student network to refine the generalization of graph representation.

**Limitations And Societal Impact:**

The authors solved some limitations but it still needs more explorations in the related area.

**Main Review:**

+ This paper proposes an interesting solution of unsupervised representation learning of the graphs. The work is solid, and it contains some novelty.
+ The experiments show that their methods were evaluated widely on 10 popular datasets and the results of the experiments showed significant improvement with the existing benchmark results.


This paper contains some uncleared descriptions and concerns. There are a few questions as listed below:

	1. Why do you add the perturbations to hidden layer 1 instead of other layers or the generated representation? Is there any discussion or comparison about this choice?
	2. Eq3. is probably confused to readers. Could you provide more details?
	3. In L 125, the definition of G and G' are unclear. What is G and G' exactly?
	4. For the experiments in Table 3, why do you compare your unsupervised learning result with supervised and kernel results?
	5. Table 4 shows some inferior results on some datasets. Is there any possible analysis and reasons?


**Time Spent Reviewing:**

3

---

> ### Author Response · Authors · 2021-08-10
> **Response to Reviewer ry4V**
>
> Thank you for the positive and constructive comments.
> We appreciate that you consider our work solid and contains some novelty.
> Below please find the responses to some specific comments.
>
> ----
>
> Q1. Why do you add the perturbations to hidden layer 1 instead of other layers or the generated representation? Is there any discussion or comparison about this choice?
>
> A1. Perturbations can be added to feature X in the input layer or the hidden layers.
> We chose to apply the perturbation at the first hidden layer.
>
> - In Table 6 on page 9, we experiment with GNN encoders for 2-5 layers  and show that using a 2-layer encoder works best. Using a 2-layer encoder requires less memory space.
>
> - We have also conducted experiments for the locations where perturbations are applied on the 4 OGB datasets. The results are shown in the table below.
> The performance of our algorithm is better when using the 2-layer encoder. Our method achieved the best empirical performance by applying the perturbation at the first hidden layer.
>
> - As the number of layers increases, we observe a decreasing trend in the algorithm's performance. A possible reason is the oversmoothing of the GCN, which causes the representation of the nodes to converge to the same.
>
> - The average number of nodes for the four OGB datasets was 25.51, 18.57, 18.78, and 24.06, respectively. GCN might be more oversmoothed on small-scale graphs.
>
>
> X: input feature
> H0: first hidden layer
> H1: second hidden layer
>
> Dataset: molHIV
>
> | #layers | perturbed | ROC-AUC(%) | #layers | perturbed | ROC-AUC(%) | #layers | perturbed | ROC-AUC(%) |
> | ---- | ---- | ---- | ---- | ---- | ---- | ---- | ---- | ---- |
> | 2    | X    | **78.7** |  3   | X    | 71.0 |  4   | X    | 72.3 |
> | 2    | H0   | **78.7** |  3   | H0   | 75.8 |  4   | H0   | 72.4 |
> | 2    | H1   | 74.6 |  3   | H1   | 76.0 |  4   | H1   | 73.2 |
> |      |      |      |  3   | H2   | 75.4 |  4   | H2   | 73.9 |
> |      |      |      |      |      |      |  4   | H3   | 71.4 |
>
> Dataset: molTox21
>
> | #layers | perturbed | ROC-AUC(%) | #layers | perturbed | ROC-AUC(%) | #layers | perturbed | ROC-AUC(%) |
> | ---- | ---- | ---- | ---- | ---- | ---- | ---- | ---- | ---- |
> | 2    | X    | 74.6 |  3   | X    | 73.6 |  4   | X    | 73.6 |
> | 2    | H0   | **74.6** |  3   | H0   | 73.2 |  4   | H0   | 74.0 |
> | 2    | H1   | 73.3 |  3   | H1   | 73.1 |  4   | H1   | 73.4 |
> |      |      |      |  3   | H2   | 72.7 |  4   | H2   | 72.8 |
> |      |      |      |      |      |      |  4   | H3   | 73.1 |
>
>
> Dataset: molToxCast
>
> | #layers | perturbed | ROC-AUC(%) | #layers | perturbed | ROC-AUC(%) | #layers | perturbed | ROC-AUC(%) |
> | ---- | ---- | ---- | ---- | ---- | ---- | ---- | ---- | ---- |
> | 2    | X    | 61.7 |  3   | X    | 59.9 |  4   | X    | 60.7 |
> | 2    | H0   | **61.9** |  3   | H0   | 60.5 |  4   | H0   | 60.9 |
> | 2    | H1   | 60.4 |  3   | H1   | 59.9 |  4   | H1   | 60.8 |
> |      |      |      |  3   | H2   | 59.6 |  4   | H2   | 60.2 |
> |      |      |      |      |      |      |  4   | H3   | 59.7 |
>
>
>
> Dataset: molBBBP
>
> | #layers | perturbed | ROC-AUC(%) | #layers | perturbed | ROC-AUC(%) | #layers | perturbed | ROC-AUC(%) |
> | ---- | ---- | ---- | ---- | ---- | ---- | ---- | ---- | ---- |
> | 2    | X    | 70.5 |  3   | X    | 67.1 |  4   | X    | 67.0 |
> | 2    | H0   | 70.6 |  3   | H0   | 69.6 |  4   | H0   | 62.9 |
> | 2    | H1   | **71.3** |  3   | H1   | 69.8 |  4   | H1   | 62.4 |
> |      |      |      |  3   | H2   | 69.8 |  4   | H2   | 62.1 |
> |      |      |      |      |      |      |  4   | H3   | 62.3 |
>
>
> ----
>
>
>
>
> Q2. Eq3. is probably confused to readers. Could you provide more details?
>
>
> A2. We apologize for the unclear presentation of Eq3.
>
> $q_{\theta} (z'_{\theta})$      is the representation of the positive view of the graph
> $z_\xi$  represents the representation of the graph G.
>
> We $\ell_2$-normalize these two representations to obtain
> $\bar{q}_{\theta} (z'_\theta)= { {q}_\theta(z'_\theta) }/{||{q}_\theta(z'_\theta) ||_2}$ and $\bar{z}_\xi = { z_\xi} / {|| z_\xi ||_2}$, meaning $ || \bar{q}_\theta(z'_\theta) ||_2 = 1$ and $ || \bar{z}_\xi ||_2 = 1$.
>
> There was a typo in Eq3 in the paper, and we correct it as follows.
>
> $$
> \mathcal{L}_{\theta,\xi}=||\bar{q}_\theta(z'_\theta)-\bar{z}_\xi||_2^2
> = || \bar{q}_\theta(z'_\theta) ||_2^2 + || \bar{z}_\xi ||_2^2 - 2\cdot \left \langle \bar{q}_\theta(z'_\theta),\bar{z}_\xi \right \rangle
> =2-2 \cdot \frac{\left \langle {q}_\theta(z'_\theta),{z}_\xi \right \rangle}{||{q}_\theta(z'_\theta)||_2 \cdot ||{z}_\xi||_2}
> $$
> ----
>
> Q3. In L 125, the definition of G and G' are unclear. What is G and G' exactly?
>
> A3. We apologize for the confusion caused by the unclear presentation of G and G'.
>
> G represents the original graph, and G' represents the perturbed graph of G.
> Here the perturbation can be applied at the input layer to feature X or at the hidden layer H of the encoder.
>
> ----
>
> Q4. For the experiments in Table 3, why do you compare your unsupervised learning result with supervised and kernel results?
>
> A4. Firstly, the comparison method we chose is for graph classification methods.
> It is common practice to compare unsupervised learning methods with supervised and kernel methods.
> Compared to supervised learning methods, our method requires less information in the training phase. Therefore, on the final graph classification task, our GASSL was able to outperform or tie the supervised learning method, validating the effectiveness of our method.
>
> -----
>
> Q5. Table 4 shows some inferior results on some datasets. Is there any possible analysis and reasons?
>
> A5. As shown in Table 4, our method does not outperform the comparison method on all data sets.
> Our method is a self-supervised learning method and does not use class labels during training.
> The comparison methods, such as the GMT method, are supervised learning methods and use class labels during training, allowing the model to better fit the dataset.
> We only train a linear classifier in the testing phase to evaluate the performance of the learned graph representation in the graph classification task.
>
> ----
>
> Finally, we sincerely thank you for your time and valuable comments.

---

> > ### Comment · Reviewer_ry4V · 2021-09-03
> > **feedback of authors response**
> >
> > Thank you for the response and more details. I think the authors have solved most of my concerns. I have updated the score accordingly.

---

> > > ### Author Response · Authors · 2021-09-04
> > > **Thank you.**
> > >
> > > We sincerely thank you for your positive comments and constructive suggestions, which have greatly helped to improve the quality of the paper.

---

### Official Review · Reviewer_9kp5 · 2021-07-22

**Rating:** 7
**Confidence:** 4

**Summary:**

This paper proposes to utilize the methodology of adversarial training to help the SSL problem in GNN. It establishes some basic formulations for the problem and validate the effectiveness on benchmark datasets.

**Main Review:**

## Pros
1) This paper is well-written. The idea is presented in a clear and straightforward manner.
2) The idea aligns with the common understandings about adversarial training. For instance, adversarial training is also applied as an data augmentation method in NLP [1].
3) The proposed method is evaluated on many benchmark datasets.

## Cons
1) Although the topic is interesting and insightful, the proposed method is somewhat primitive.
2) The improvements is relatively marginal. Sometimes the improvement is less than 0.5%, which may well be noise. The variance shown in Table 3 also suggests that the improvement is not very stable.
3) If I was right, this paper only provide evaluations on the graph classification task but omit the commonly used node classification task. Please explain.
4) The proposed method should be compared with deep graph infomax. The [44] cited paper in this work.
5) I am curious about the robustness of the proposed method. Can the author provide corresponding results [2][3]?


I may raise my score if the authors can properly answer my questions.

[1] Adversarial Mutual Information for Text Generation

[2] N. Entezari, S. A. Al-Sayouri, A. Darvishzadeh, and E. E. Papalex-akis,  “All  you  need  is  low  (rank)  defending  against  adversarialattacks on graphs,” inWSDM, 2020, pp. 169–177

[3] H.  Wu,  C.  Wang,  Y.  Tyshetskiy,  A.  Docherty,  K.  Lu,  and  L.  Zhu,“Adversarial  examples  on  graph  data:  Deep  insights  into  attackand defense,” inIJCAI, 2019


**Time Spent Reviewing:**

6

---

> ### Author Response · Authors · 2021-08-10
> **Response to Reviewer 9kp5**
>
> We sincerely thank you for your time and efforts.
> We appreciate that you find our approach clear and straightforward.
> Below please find the responses to some specific comments.
> ----
>
>
> Q1. Although the topic is interesting and insightful, the proposed method is somewhat primitive.
>
> A1. The method is simple but effective. Our method overcomes the shortcomings of manually designed views and provides a model-agnostic solution for automatically learning views. Experiments on 10 datasets show that our method has superior performance on graph classification tasks than other self-supervised methods.
>
>
> ----
>
> Q2. The improvements is relatively marginal. Sometimes the improvement is less than 0.5%, which may well be noise. The variance shown in Table 3 also suggests that the improvement is not very stable.
>
> A2. We respectfully disagree with the reviewer that the improvements are relatively marginal.
> Although the improvement over the MVGRL method is less than 0.5% on both the IMDB-B and IMDB-M datasets, however, on the majority of other datasets, our method showed consistent improvement.
> Compared to SOTA self-supervised learning methods such as InfoGraph, MVGRL, and GCC, our method has an average improvement of 3.6%, 2.6% and 5.3%, respectively.
> Compared to the GraphCL, our method improves by 4.1%, 3.1%, 6.7%, and 2.2% on the MUTAG, IMDB-B, COLLAB, and NCI1 datasets, respectively.
>
> Another issue is the relatively large variance of the algorithm performance.
> From Table 2, the variance of the classification performance of our method on the MUTAG and PTC-MR datasets is relatively large due to the small size of these two datasets.
> Specifically, the MUTAG dataset contains 188 graphs, and PTC-MR contains 344 graphs.
> 10-fold cross-validation is a common practice when validating the performance of graph classification tasks. The size of the test sets was approximately 19 and 34, respectively.
> The small size of the test set resulted in a relatively large variance.
> We observe from Table 3 that the variance of our algorithm is reduced on the larger datasets IMDB-B, IMDB-M, COLLAB, and NCI1.
>
>
> ----
> Q3. If I was right, this paper only provide evaluations on the graph classification task but omit the commonly used node classification task. Please explain.
>
> A3. Yes. Our work focus on the graph-level task.
> Our approach learns the representation of the whole graph to improve the performance of downstream tasks, i.e., graph classification.
> As our approach is not focused on node-level tasks, experiments on node classification are not in the scope of this paper.
> However, we can experiment on node classification task, it requires some modification of the objective function.
> We report some results in A4 to Q4.
>
> ----
>
> Q4. The proposed method should be compared with deep graph infomax. The [44] cited paper in this work.
>
> A4. Following the suggestions and requests of the reviewers, we experimented with our method on the node classification task.
> We used the same experimental setup as Deep Graph Infomax (DGI).
> We perturb all the nodes and modify the loss function.
> We modify the loss function. In specific, we take the sum of the distances between the original representation and the perturbed representation of each node as the loss.
> We kept the other algorithm settings unchanged and trained it in a self-supervised learning approach, using the BYOL framework, and named GASSL_node.
> We run 10 times experiments and report the average test accuracy.
>
> | Method | Cora  | Citeseer | Pumbed |
> | ----   | ----  | ---- | ---- |
> | DGI    | 82.3  | 71.8 | 76.8 |
> | GCN    | 81.5  | 70.3 | 79.0 |
> | GASSL_node(ours) | 76.8| 65.0| 73.9 |
> | GCN_adv (ours) | 82.7 | 71.9 | 80.8 |
>
> Our method performs worse than DGI and GCN on all three datasets.
> The possible reason for this is that GASSL_node ignores the relationship between the node representation and the global representation.
>
> Due to time constraints, we did not continue to optimize the original self-supervised model.
> We made a new attempt on the GCN, using the semi-supervised GCN as a basis for adversarial perturbation of all nodes, named GCN_adv.
> As can be seen from the table, this method outperforms DGI and GCN in terms of accuracy.
> The experimental results show that adversarial perturbation can improve the model accuracy on the node classification task.
> However, since our approach focuses mainly on the graph classification task, we leave the node classification task as future work.
>
> ----
> Q5. I am curious about the robustness of the proposed method. Can the author provide corresponding results [2][3]?
>
> A5. Robustness has always been an interesting topic in GNN research. However, it might be outside the scope of this paper.
> Due to time and experimental constraints, we could not complete the robustness experiments within a week, and we will continue to explore the robustness of the proposed method in future versions.
> We are glad to report on the robustness performance of our method once the experiments are completed.
>
> ----
>
> We are sorry that due to time constraints we were not able to complete all the experiments.
> Thank you for your very helpful and constructive review comments.

---

> > ### Comment · Reviewer_9kp5 · 2021-08-20
> > **Thanks for your reply.**
> >
> > The rebuttal generally solves my concerns, except for Q3&4.
> >
> > In the rebuttal, the authors confirm that their method can not be applied to the very important task of node classification. However, with some adaptation, the method can achieve improvement (GCN_adv). They also claim that this adaptation will not be added to the main content for now. The explanation of the adaptation (using the semi-supervised GCN as a basis for adversarial perturbation of all nodes) is somewhat vague and unclear to me. Thus, I believe the current version of this work does have some contribution, but meanwhile the contribution is limited to a specific field and is not very universal. Only if the author can provide a complete and clear examination on the important node classification task will I raise my score to 7. Otherwise, I tend to keep my score to 6. I suggest the author to provide an anonymous link to a pdf file, which should introduce GCN_adv in detail and a full evaluation of the node classification tasks. Also, this content should be added to the main content of the paper for the sake of helping future works.

---

> > > ### Author Response · Authors · 2021-08-23
> > > **Response to the node classification task.**
> > >
> > > Thank you for your positive feedback on our work and for your constructive comments.
> > >
> > > According to the reviewers' request, we have analysed and experimented the GASSL method on the node classification task. Two strategies are proposed: a two-stage training approach and a joint training approach. For details, please see
> > >
> > > https://github.com/Anonymous20201007/losscurves/blob/main/response.pdf
> > >
> > > We hope that our response will address your concerns.

---

> > > > ### Comment · Reviewer_9kp5 · 2021-08-27
> > > > **Thanks for the reply**
> > > >
> > > > I am satisfied with the extra pdf file and thus raise my score. Please keep in mind adding this extra part to the camera-ready.

---

> > > > > ### Author Response · Authors · 2021-08-28
> > > > > **Thank you.**
> > > > >
> > > > > We sincerely appreciate the reviewer for the great suggestions and insights, which enabled us to further strengthen our work and make our paper stronger. We will add the extra part to the final version.

---

### Official Review · Reviewer_rGkH · 2021-07-23

**Rating:** 6
**Confidence:** 4

**Summary:**

The paper investigates the unsupervised representation of graph data and proposes an adversarial self-supervised learning (GASSL) framework. The paper adopts the self-supervised framework in reference [12] for graph embedding representation, and apply adversarial learning to improve the robustness of the model, following the framework in [26]. A number of experimental results are given to validate the performance of the proposed model. The contribution of the paper is incremental.

**Limitations And Societal Impact:**

I would like to suggest that the authors provide to visualize the feature distribution of graph representation to validate results.
Some minor comments:
L84-87. Some paragraphs need to be rewritten. The paper states " For a thorough review, we refer the reader to the recent survey [50] ", and then gives two references later.
L104.  "[26] proposed " ==> "Authors [26]  proposed "


**Main Review:**

The problem addressed in the paper is significant and interesting to the readers of the conference. The main contribution of the paper is to extend the self-supervised learning framework to graph data and to adopt adversarial learning to improve the robustness of the model. Both self-supervised learning framework and adversarial learning model used in the paper are not original and available in the reference. The novelty of the paper is to combine these two models to learn a robust graph representation.
The paper provides a number of experimental results to support that good performance of the proposed approach and ablations experimental results are also given, but it lacks of sufficient evidence that why the proposed method performs well. The reviewer did not see implications from experimental results. In particular, the paper claims that “We use adversarial training to automatically generate challenging views for self-supervised learning in place of handcrafted views, which yield performance gains on multiple datasets.”  From the procedure of adversarial generator, the model is to generate an adversarial attack in the vicinity of the input sample that maximizes its loss function, how to guarantee that the generated sample should be the different view of the same instance? What is the definition of "challenging views".
English writing is clear and easy to understand.

**Time Spent Reviewing:**

30

---

> ### Author Response · Authors · 2021-08-10
> **Response to Reviewer rGkH**
>
> Thank you for the detailed and constructive comments.
> Below please find the responses to some specific comments.
> ----
>
> Q1. The problem addressed in the paper is significant and interesting to the readers of the conference. The main contribution of the paper is to extend the self-supervised learning framework to graph data and to adopt adversarial learning to improve the robustness of the model. Both self-supervised learning framework and adversarial learning model used in the paper are not original and available in the reference. The novelty of the paper is to combine these two models to learn a robust graph representation.
>
> A1. Thank you for pointing out that our paper is addressing an important and interesting problem.
> Adversarial learning is commonly used to improve model robustness; however, this is not the main focus of this paper. This paper focuses on how to automatically construct challenging views, thus avoiding the effort and trial and error of hand-designed views and improving graph self-supervised representation learning performance in downstream tasks.
> We use adversarial learning to automatically construct challenging views, providing an effective solution for learning whole graph representations in a self-supervised learning framework.
>
> ----
>
> Q2. The paper provides a number of experimental results to support that good performance of the proposed approach and ablations experimental results are also given, but it lacks of sufficient evidence that why the proposed method performs well. The reviewer did not see implications from experimental results. In particular, the paper claims that “We use adversarial training to automatically generate challenging views for self-supervised learning in place of handcrafted views, which yield performance gains on multiple datasets.” From the procedure of adversarial generator, the model is to generate an adversarial attack in the vicinity of the input sample that maximizes its loss function, how to guarantee that the generated sample should be the different view of the same instance?
>
> A2. The problem of guaranteeing that the generated samples are different views of the same instance is also faced by all hand-designed view methods.
> We empirically believe that the key lies in the radius of perturbation and the separation of the dataset.
> Inspired by [1], we analyze the separation of the OGB dataset.
> We use the mean-pooing of initial node features as a representation of the graph.
>
> Dataset | Train-Train Separation | Test-Train Separation
> ---- | ---- | ----
> BBBP | 0.1604 | 0.1259 |
> HIV  | 0.1020 | 0.1138 |
> Tox21| 0.0719 | 0.0811 |
> ToxCast | 0.0912 | 0.0983 |
>
> The Train-Train Separation is the $\ell_{\infty} $ distance between each training example
> and its closest neighbor with a different class label in the training set, while the Test-Train Separation is the $\ell_{\infty} $ distance between each test example and its closest example with a different class in the training set.
> [1] show that it is theoretically possible to achieve both robustness and accuracy for r-separated data.
>
> Observation: For the BBBP dataset, the Train-Train Separation and Test-Train Separation are 0.1604 and 0.1259, respectively. We observed in Figure 2 of the paper that the clean accuracy decreases as the perturbation radius increases. In particular, when the perturbation radius is larger than the BBBP separation, the clean accuracy decreases significantly.
>
> In this paper, we use a perturbation radius of 8e-3, which is much smaller than r.
>
> [1] Yang, Yao-Yuan, et al. “A Closer Look at Accuracy vs. Robustness.” Advances in Neural Information Processing Systems, vol. 33, 2020, pp. 8588–8601.
>
> ----
>
> Q3. What is the definition of "challenging views".
>
> A3. The challenging view is the one that causes the encoder to produce a new view that is significantly different from the representation of the original sample.
> We use adversarial learning to generate challenging views within a certain perturbation radius.
>
> ----
>
> Q4. I would like to suggest that the authors provide to visualize the feature distribution of graph representation to validate results.
>
> A4. Thank you very much for your suggestion. We have mapped the resulting graph representation onto a sphere to give a more precise illustration.
>
> ----
>
> Q5. L84-87. Some paragraphs need to be rewritten. The paper states " For a thorough review, we refer the reader to the recent survey [50] ", and then gives two references later.
>
> A5. We are very sorry for the confusion caused by our writing. We will rewrite this paragraph.
>
> ----
>
> Q6. L104. "[26] proposed " ==> "Authors [26] proposed "
>
> A6. Thank you very much for your comments. We accept and amend them accordingly.
>
> ----
>
> Finally, thank you very much for your detailed review comments, which have been extremely helpful and enlightening.

---

### Decision · Program_Chairs · 2021-09-27

**Decision:**

Accept (Poster)

**Comment:**

This paper investigates the problem of self-supervised learning in the context of graph representation learning. It proposes to adopt the technique of adversarial training to automatically augment a training set, and then devises the corresponding adaptation scheme to make adversarial training viable. The authors conducted thorough and insightful experiments (also supplementing some experimental results in the rebuttal) on several benchmark datasets.

Although adversarial training was originally proposed as a defensive algorithm aiming at increasing the robustness of a certain learning model, it has also been recognized as a variant of data augmentation or hard example mining in many other works. Moreover, employing the philosophy of adversarial training to help generalization has also been discussed in other fields like NLP and style transformation. Thus, it is reasonable to see that the same theory can be verified in the domain of graph learning. Of course, verifying this theory in a new domain like graph learning requires huge efforts and insightful designs, which holds as the main contribution of this work. The proposed method is relatively simple but very effective on the evaluated tasks. In the rebuttal phase, the authors adequately answered the questions from the reviewers, including addressing the issues of writing clarity and evaluations in extra experimental settings. Eventually, the four reviewers reached the consensus of accepting the paper. Therefore, the AC recommends acceptance as poster regarding this submission.